# The Prolyl Oligopeptidase Inhibitor KYP-2047 Is Cytoprotective and Anti-Inflammatory in Human Retinal Pigment Epithelial Cells with Defective Proteasomal Clearance

**DOI:** 10.3390/antiox12061279

**Published:** 2023-06-15

**Authors:** Maija Toppila, Maria Hytti, Eveliina Korhonen, Sofia Ranta-aho, Niina Harju, Markus M. Forsberg, Kai Kaarniranta, Aaro Jalkanen, Anu Kauppinen

**Affiliations:** 1School of Pharmacy, Faculty of Health Sciences, University of Eastern Finland, 70211 Kuopio, Finland; 2Department of Ophthalmology, Institute of Clinical Medicine, University of Eastern Finland, 70211 Kuopio, Finland; 3Department of Clinical Chemistry, University of Helsinki and Helsinki University Hospital, 00014 Helsinki, Finland; 4Department of Ophthalmology, Kuopio University Hospital, 70211 Kuopio, Finland

**Keywords:** prolyl oligopeptidase inhibitor, retinal pigment epithelium, age-related macular degeneration, proteasomal dysfunction, ROS, inflammation, MAPKs

## Abstract

Increased oxidative stress, dysfunctional cellular clearance, and chronic inflammation are associated with age-related macular degeneration (AMD). Prolyl oligopeptidase (PREP) is a serine protease that has numerous cellular functions, including the regulation of oxidative stress, protein aggregation, and inflammation. PREP inhibition by KYP-2047 (4-phenylbutanoyl-L-prolyl1(S)-cyanopyrrolidine) has been associated with clearance of cellular protein aggregates and reduced oxidative stress and inflammation. Here, we studied the effects of KYP-2047 on inflammation, oxidative stress, cell viability, and autophagy in human retinal pigment epithelium (RPE) cells with reduced proteasomal clearance. MG-132-mediated proteasomal inhibition in ARPE-19 cells was used to model declined proteasomal clearance in the RPEs of AMD patients. Cell viability was assessed using LDH and MTT assays. The amounts of reactive oxygen species (ROS) were measured using 2′,7′-dichlorofluorescin diacetate (H2DCFDA). ELISA was used to determine the levels of cytokines and activated mitogen-activated protein kinases. The autophagy markers p62/SQSTM1 and LC3 were measured with the western blot method. MG-132 induced LDH leakage and increased ROS production in the ARPE-19 cells, and KYP-2047 reduced MG-132-induced LDH leakage. Production of the proinflammatory cytokine IL-6 was concurrently alleviated by KYP-2047 when compared with cells treated only with MG-132. KYP-2047 had no effect on autophagy in the RPE cells, but the phosphorylation levels of p38 and ERK1/2 were elevated upon KYP-2047 exposure, and the inhibition of p38 prevented the anti-inflammatory actions of KYP-2047. KYP-2047 showed cytoprotective and anti-inflammatory effects on RPE cells suffering from MG-132-induced proteasomal inhibition.

## 1. Introduction

Prolyl oligopeptidase (PREP), also called prolyl endopeptidase with even more abbreviations (e.g., PO, PE, PEP, or POP), is a serine protease with the ability to hydrolyze peptides smaller than 30 amino acids [1]. It plays a role in the regulation of memory and learning, α-synuclein aggregation, neuronal processes, angiogenesis, and the regulation of oxidative stress and inflammation [2,3,4,5,6,7]. There is convincing evidence that PREP activity is altered in certain inflammatory conditions. For example, PREP activity increases in patients with cystic fibrosis or rheumatoid arthritis but decreases in patients suffering from multiple sclerosis [8,9,10].

The diverse functions of PREP are mediated either by its peptidase activity or by direct protein-protein interactions and seem to vary depending on both the cell type and on its localization (i.e., either inside or outside of the cell) [10,11]. KYP-2047 (4-phenylbutanoyl-L-prolyl1(S)-cyanopyrrolidine) is a PREP inhibitor that causes a conformational stabilization of PREP’s active site and thus regulates protein-protein interactions leading to, for example, autophagy activation, reduced oxidative stress, or the regulation of inflammation [7,12].

PREP activity and expression also increase during aging in various tissues. It has been postulated that this increase is related to an accumulation of oxidative stress and toxic protein aggregates with aging, and PREP inhibition might have therapeutic implications in some neurodegenerative diseases [13,14].

Aging is also a major risk factor for age-related macular degeneration (AMD), a progressive eye disease causing blindness and diminished quality of life among the elderly. Aging, chronic oxidative stress, and protein accumulation impair the retinal pigment epithelium (RPE) cell layer, eventually leading to cell death. RPE cells perform several functions that ensure normal visual perception, including maintenance of the photoreceptor cells. The loss of RPE cells in the macula during AMD secondarily triggers the death of photoreceptors and subsequent central blindness, causing difficulties in reading and facial recognition [15,16,17]. The key factors underlying AMD are chronically increased oxidative stress and impaired cellular clearance mechanisms, which lead to the accumulation of protein aggregates, cellular waste products, and damaged organelles within the cell. This initiates a maladapted inflammatory response that causes RPE cell death and ultimately AMD.

PREP inhibition has been shown to have significant potential in clearing aggregated proteins such as α-synuclein and being able to reduce the level of reactive oxygen species (ROS) and inflammation [7,18], which makes PREP inhibition an interesting therapeutic approach for AMD. In the present study, we investigate the effects of KYP-2047 on inflammation, cell viability, oxidative stress, and autophagy in human RPE cells with reduced proteasomal clearance.

## 2. Materials and Methods

### 2.1. Cell Culture

The human retinal pigment epithelium cell line ARPE-19 (passage 19) was purchased from the American Type Culture Collection (ATCC, Manassas, VA, USA) and cultured in a 1:1 mixture of Dulbecco’s modified Eagle’s medium (DMEM) and nutrient mixture F-12 (Life Technologies, Carlsbad, CA, USA). The medium was supplemented with 10% HyClone fetal bovine serum (FBS) (Thermo Fisher Scientific, Waltham, MA, USA), 100 U/mL of penicillin, 100 μg/mL of streptomycin (Lonza, Basel, Switzerland or Life Technologies, Carlsbad, CA, USA for both), and 2 mM of L-glutamine (Life Technologies, Carlsbad, CA, USA). The cells were passaged every 3–4 days using 0.25% Trypsin–EDTA (Life Technologies, Carlsbad, CA, USA) and used for experiments in passages between 26 and 35. The cells were maintained routinely and during experiments in an incubator providing a humidified atmosphere at +37 °C with 5% CO_2_.

### 2.2. Cell Treatments

For the experiments, the cells were plated on 12-well plates at a density of 200,000 cells/well or 96-well plates at a density of 20,000 cells/well (ROS measurements) and cultured for 3 days to reach confluency. The confluent cells were washed with serum-free DMEM/F12 medium. The cells were exposed to 0.1–10 µM KYP-2047 (synthesized by Dr. Elina Jarho, School of Pharmacy, UEF, Finland) for 15 min in a serum-free culture medium either with or without initial 24 h priming with human recombinant interleukin-1α (IL-1α, 4 ng ml−1, R&D Systems, Abington, UK) in a serum-free medium. Next, proteasome inhibition was induced by exposing the cells to MG-132 (5 µM, Calbiochem, San Diego, CA, USA) for an additional 2 h, 6 h, 24 h, or 48 h. The experiments were also performed with specific inhibitors against p38 mitogen-activated protein kinase (MAPK; 50 μM, SB203580, Cell Signaling Technologies, Beverly, MA, USA) or ERK1/2 (50 μM, PD98059, Cell Signaling Technologies, Beverly, MA, USA) under the same conditions. KYP-2047, MG-132, and both MAPK inhibitors were dissolved in dimethyl sulfoxide (DMSO, Sigma-Aldrich, St. Louis, MO, USA), and the vehicle controls were exposed to identical DMSO concentrations.

### 2.3. Cell Viability Assays

A lactate dehydrogenase (LDH, CytoTox96^®^ Non-Radioactive Cytotoxicity Assay, Promega, Madison, WI, USA) assay was used to assess the integrity of the cell membrane. The LDH levels were measured from fresh medium samples according to the manufacturer’s protocol. In addition, the metabolic activity of the cells was determined using a 3-(4,5-dimethylthiazol-2-yl)-2,5-diphenyltetrazolium bromide (MTT, Sigma-Aldrich, St Louis, MO, USA) assay. Briefly, an MTT salt solution was added to the cell culture medium at a final concentration of 500 µg/mL, and the cells were incubated for 1.5 h in an incubator providing a humified atmosphere at +37 °C with 5% CO_2_ and protection from light. After incubation, the medium was removed and replaced with 1 mL DMSO (Fischer Scientific, Leics, UK), which was incubated for 20 min at room temperature (RT) to facilitate the dissolution of formazan crystals. Subsequently, 200 µL of the dissolved formazan solution in DMSO from each well was transferred onto a 96-well plate, and the absorbance values were measured using a spectrophotometer (BioTek, ELx808, Microplate reader with the Gen-5 2.04 program, BioTek Instruments Inc, Winooski, VT, USA) at a wavelength of 562 nm.

### 2.4. ROS Detection

The ROS-sensitive fluorescent dye 2′,7′-dichlorodihydrofluorescein diacetate (H2DCFDA, Life technologies, Eugene, OR, USA) was used to determine the ROS levels in the ARPE-19 cells. After IL-1α priming, the cells were exposed to 5 µM H2DCFDA and 1 µM KYP-2047 for 15 min prior to the addition of 5 µM MG-132 for 45 min. The vehicle controls were exposed to identical DMSO concentrations. After incubation, the cells were washed twice with Dulbecco’s Phosphate Buffered Saline (DPBS, Life Technologies, Paisley, UK), and fresh DPBS was added. The fluorescence intensity (ex/em = 485/530 nm) was measured using a BioTek Cytation3 imaging reader (BioTek, Instruments Inc, Winooski, VT, USA).

### 2.5. PREP Activity Measurement

PREP activity was measured from the cell lysate samples according to the protocol published by Natunen et al. [18]. In this procedure, 0.8 mM Suc-Gly-Pro-7-Amino-4-methylcoumarin (Suc-Gly-Pro-AMC; Bachem AG, Basel, Switzerland) was used as a substrate for PREP, and the PREP levels were determined using a Victor2 fluorescence plate reader (PerkinElmer Inc., Waltham, MA, USA).

### 2.6. Enzyme-Linked Immunosorbent Assay (ELISA)

ELISA was used to determine the pro-inflammatory cytokine IL-6, IL-8, and IL-1β levels from the cell culture medium samples. BD OptEIA^™^ Human ELISA Kits (BD Biosciences, San Diego, CA, USA) were used according to the manufacturer’s protocol. The absorbances were measured in a spectrophotometer at a wavelength of 450 nm, with a correction wavelength of 620 nm. For the levels of phosphorylated MAPKs p38, c-Jun N-terminal kinase (JNK), and extracellular signal-regulated kinases (ERKs), half were determined from whole cell lysates using human PathScan^®^ ELISA Kits obtained from Cell Signaling Technologies (Beverly, MA, USA) according to the manufacturer’s instructions.

### 2.7. Western Blot

Autophagy markers p62/SQSTM1 and LC3 were detected using the western blot method. The cells were lysed with M-PER (Thermo Fisher Scientific, Waltham, MA, USA) according to the manufacturer´s instructions, and the protein levels of the collected lysates were measured using the Bradford protocol [19]. Equal amounts of protein (30 µg/sample) were loaded onto a 15% SDS-PAGE gel, and the protein bands were separated at 200 V. The samples were wet-blotted onto a nitrocellulose membrane (GE Healthcare, Little Chalfont, Buckinghamshire, UK) overnight at 17 V with a blotting buffer containing 20% methanol. Ponceau S staining (Sigma-Aldrich, St. Louis, MO, USA) was used to confirm the protein transfer, and thereafter, the membrane was blocked with 3% milk in 0.3% TWEEN-20/PBS for 1.5 h at RT. The primary antibodies for p62/SQSTM1 (1:1000 in 0.5% BSA in 0.3% tween-PBS; sc-28359; Santa Cruz Biotechnology, Santa Cruz, CA, USA) and LC3 (1:1000 in 0.1% TWEEN-20 1 x TBS; AP1802a; Abgent, San Diego, CA, USA) were incubated overnight at +4 °C. The primary antibody for GAPDH (1:15,000 in 0.1% T-PBS; ab8245; Abcam, Cambridge, UK) was incubated for 2 h at RT. After washing for 3 × 5 min with the respective washing buffer, the membranes were incubated with secondary antibodies. The anti-mouse secondary antibody for p62/SQSTM1 (NA931; GE Healthcare, Chicago, IL, USA) was incubated for 2 h at RT (1:10,000 in 3% milk in 0.3% TWEEN-20/PBS) and for 1 h at RT for GAPDH (1:12,000 in 0.1% TWEEN-20/PBS). The anti-rabbit secondary antibody for LC3 (1:5000 in 3% milk/0.1% TWEEN-20/TBS; A16104; Invitrogen, Carlsbad, CA, USA) was incubated for 2 h at RT. The washing steps were repeated before exposure to the substrate (Millipore, Billerica, MA, USA) and subsequent detection with an Image Quant RT ECL-camera (GE Healthcare, Little Chalfont, UK) or X-ray film (Fuji Corporation, Tokyo, Japan). The relative densities of the bands from the images or scanned films were quantified using ImageJ software (US National Institutes of Health, Bethesda, MD, USA; http://imagej.nih.gov/ij/ (accessed on 14 March 2023) and normalized to GAPDH.

### 2.8. Statistical Analysis

Statistical analyses were performed with GraphPad Prism version 8 (Graphpad Software, San Diego, CA, USA). Pairwise comparisons were performed using the Mann–Whitney U test, and a *p* value < 0.05 was considered statistically significant.

## 3. Results

### 3.1. PREP Inhibitor KYP-2047 Was Tolerated Well by the ARPE-19 Cells

In order to study the tolerability of KYP-2047 in human ARPE-19 cells, the cells were exposed to different concentrations of the PREP inhibitor KYP-2047. According to ISO standard 10993-5, viability should reach 80% or more in viability assays [20]. In the MTT assay, cell viability exceeded 90% at all measured KYP-2047 concentrations (0.1 µM = 93%; 1 µM = 94%; 5 µM = 95%; and 10 µM = 92%), indicating that the cells remained viable upon exposure to KYP-2047 at least up to a 10 µM concentration, although viability was significantly reduced at the 0.1 and 10 µM KYP-2047 concentrations in comparison with the untreated control cells (Figure 1A). LDH leakage significantly increased at 1 µM, while no other concentration had elevated LDH levels in the cell culture medium (Figure 1B). Microscopy images of the cells further supported good tolerability, and the cells exposed to all of the studied concentrations of KYP-2047 were visibly similar to the control cells (Figure 1C).

### 3.2. PREP Inhibitor KYP-2047 Reduced the Production of Pro-Inflammatory Cytokines and Protected the ARPE-19 Cells from MG-132-Induced Cytotoxicity

Next, we measured the levels of LDH and inflammatory cytokines IL-6, IL-8, and IL-1β. The ARPE-19 cells were exposed to a 5 µM concentration of the proteasome inhibitor MG-132 for 24 h, according to our previously established protocol [21]. MG-132 significantly increased the leakage of LDH from the cells. KYP-2047 pretreatment reduced LDH leakage by 31.9% (Figure 2A). Additionally, exposure to MG-132 lowered the levels of IL-6, and 1 µM KYP-2047 significantly reduced them further in the ARPE-19 cells (Figure 2B). Lower (0.1 µM) or higher (5 µM) concentrations of the PREP inhibitor had no significant effects (Figure 2B). MG-132 significantly increased the levels of IL-8 (Figure 2C), and pretreatment of the cells with 0.1 or 1 µM KYP-2047 tended to reduce the IL-8 levels, but the effect did not reach statistical significance (Figure 2C, *p* = 0.0664 for 1µM KYP-2047). The highest KYP-2047 concentration did not cause any significant change in comparison with the MG-132-treated cells (Figure 2C). In order to measure the release of the inflammasome-dependent IL-1β cytokine, a priming signal was given by pretreating the cells with IL-1α for 24 h before the 48 h of MG-132 exposure. Similar to IL-8, a subsequent exposure to MG-132 significantly increased the IL-1β release, and an exposure to 1 µM KYP-2047 tended to prevent it (Figure 2D, *p* = 0.0575). The results were similar to those with 5 µM KYP-2047, whereas 0.1 µM of the PREP inhibitor significantly increased IL-1β production (Figure 2D, *p* = 0.0997 and *p* = 0.003, respectively). On the basis of the common trend, 1 µM KYP-2047 was selected for more detailed examination in this study. Together, the cell viability, IL-6, IL-8, and IL-1β data suggest that the PREP inhibitor KYP-2047 has protective potential and anti-inflammatory properties at macromolar concentrations in ARPE-19 cells suffering from dysfunctional protein clearance.

### 3.3. KYP-2047 Inhibited Prolyl Oligopeptidase Activity in the ARPE-19 Cells

In order to verify the ability of KYP-2047 to regulate PREP in ARPE-19 cells, we measured the intracellular PREP activity. Neither IL-1α priming nor MG-132 had any effect on the intracellular PREP activity (Figure 3). Instead, 1 µM KYP-2047 significantly reduced the PREP activity to a level of 70% of the control (control = 100%; IL-1α = 114%; IL-1α + MG-132 = 114%; and IL-1α + KYP-1047 + MG-132 = 70%; Figure 3).

### 3.4. MG-132 Increased ROS Production in ARPE-19, and KYP-2047 Tended to Prevent It

In order to study the ROS production, we measured the fluorescence intensity after 1 h of exposure to a H2DCFDA probe. Proteasomal inhibition induced by 5 µM MG-132 significantly increased ROS production (Figure 4), which was reduced by exposure to 1 µM KYP-2047 (*p* = 0.09).

### 3.5. KYP-2047 Did Not Induce Autophagy in ARPE-19 Cells upon Exposure to MG-132

Previous studies on PREP inhibition in Parkinson’s disease models found some evidence for the activation of autophagy [22]. Here, we measured the expression levels of two autophagy markers, p62/SQSTM1 and LC3, using the western blot method (Figure 5A). The exposure of ARPE-19 cells to MG-132 for 6 h did not change the autophagy substrate p62/SQSTM1 levels when compared with the vehicle control (Figure 5B). KYP-2047 tended to increase the levels of p62/SQSTM1 in the MG-132-treated cells, but the effect was not quite statistically significant (*p =* 0.0649, Figure 5B). Proteasome inhibition by MG-132 induced the lipidation of LC3. KYP-2047 had no additional effect on the conversion of LC3-I to LC3-II (Figure 5 C,D).

### 3.6. Exposure of ARPE-19 Cells to KYP-2047 Led to Phosphorylation of MAPK p38 and ERK1/2

In order to determine the mechanism behind the reduced IL-6 release, we measured the level of phosphorylation of the MAPKs after 2 h of exposure of the ARPE-19 cells to KYP-2047 and MG-132. Neither MG-132 nor KYP-2047 exerted an effect on the phosphorylation of JNK, suggesting that KYP-2047 was not functioning via this pathway (Figure 6A). MG-132 tended to reduce the phosphorylation of ERK1/2 (p44/42), but this trend was not statistically significant (*p =* 0.0931). In contrast, treatment with the PREP inhibitor KYP-2047 increased the phosphorylation of ERK1/2 by 22% when compared with the MG-132 group (Figure 6B). Exposure of ARPE-19 to MG-132 significantly increased the level of phosphorylated p38, and KYP-2047 treatment increased it further by 32% (Figure 6C). Taken together, our results suggest that the PREP inhibitor KYP-2047 increases the phosphorylation of MAP kinases ERK1/2 and p38.

### 3.7. p38 Regulated the Anti-Inflammatory Effects of KYP-2047

To verify the role of MAPKs in the cytoprotective and anti-inflammatory effects of KYP-2047, we measured the levels of LDH, IL-6, and IL-8 in the presence of p38 and ERK1/2 inhibitors (Figure 7). The inhibitors did not have any effect on LDH leakage from the cells into the medium (Figure 7A). Inhibition of p38 resulted in increased levels of IL-6 (46.7%, Figure 7B) and IL-8 (23.8%, Figure 7C), implying that the anti-inflammatory effect of KYP-2047 was mediated via p38. Inhibition of ERK1/2 decreased the IL-8 levels, suggesting that the observed increase in phosphorylation of ERK1/2 seen after KYP-2047 exposure (Figure 6B) was not involved in the anti-inflammatory pathway altered by KYP-2047.

## 4. Discussion

Aging and chronic oxidative stress reduce the functionality of cellular clearance processes in the RPE. In turn, this leads to the accumulation of protein aggregates, cellular waste products, and damaged organelles inside the cells, which initiate an inflammatory response, ultimately contributing to cell death [15,17,23]. In this study, we simulated these conditions in the ARPE-19 cell line by inhibiting proteasomal clearance with MG-132. ARPE-19 is a spontaneously arising human retinal pigment epithelium cell line that is in standard use in age-related macular degeneration research. The cells express no mesenchymal markers at the passages used in this study (unpublished data). The aim of this research was to study the effects of PREP inhibitor KYP-2047 on inflammation and autophagy in retinal pigment epithelial cells.

PREP is expressed ubiquitously in the body, including in the RPE cell layer [24], which is in line with our results demonstrating the presence of PREP activity in ARPE-19 cells. Here, KYP-2047 reduced PREP activity, while proteasome inhibition had no effect on it. We preferred to determine the enzyme activity rather than the PREP protein levels since PREP activity and protein expression levels do not necessarily correlate, as shown recently by Hellinen et al. [25]. PREP functions both via its peptidase activity and protein-protein interactions, and the latter does not require the inhibition of PREP’s catalytic activity [12]. Therefore, it is possible that the observed protective and anti-inflammatory effects in our cell model were not proportional to the inhibitory effects on PREP’s catalytic activity but resulted from protein-protein interactions. PREP inhibitors have been shown to enhance the clearance of α-synuclein aggregates via autophagy activation and to reduce oxidative stress in cellular and animal models of Parkinson’s disease as well as in ARPE-19 cells with defunct proteasome activity [5,7,22,25,26]. These effects were most likely facilitated by increased activity of protein phosphatase 2 (PP2A). Active PREP inhibits PP2A activity by regulating the interactions between its subunit and regulatory proteins. The PREP inhibitor KYP-2047 facilitates the activation of PP2A downstream kinases such as death-associated protein kinase 1 (DAPK1) and beclin-1, which subsequently leads to the induction of autophagy and a reduction in ROS [7,22].

In our study, proteasome inhibition caused rupturing of the cell membrane, which was evident as elevated LDH levels in the cell culture media, a result in line with our group’s previous findings [21]. Exposure of MG-132-treated ARPE-19 cells to 1 µM KYP-2047 reduced LDH leakage, indicative of a protective effect of the PREP inhibitor on the cell membrane integrity and cell survival. Natunen et al. demonstrated that KYP-2047 protected the mouse’s primary neurons from microglial toxicity [18]. Similar cytoprotective effects have also been observed in a human neuroblastoma cell line overexpressing α-synuclein [26] and in Huntington’s disease-simulating HeLa cells [27]. These studies suggested that PREP inhibition increases autophagy, which in turn reduces the amounts of toxic proteins and improves cellular survival [18,26,27]. While we also observed higher cell viability, our present data did not support the concept that there was an activation of KYP-2047-mediated autophagy in the ARPE-19 cells. It should be noted that MG-132 inhibited proteasomal clearance, leading to accumulation of the autophagy substrate p62/SQSTM1 and increased lipidation of LC3 in ARPE-19 cells [16], as observed in our study. This might have obscured any action of KYP-2047 toward autophagy in our model.

Our data indicate that dysfunctional cellular clearance via MG-132 exposure increased ROS production. This is in line with our previous data showing that oxidative stress induced by MG-132 is a key factor in the consequent NLRP3 inflammasome activation [28]. Previous studies on PREP inhibitors showed reduced ROS production by PP2A activation in different cell lines [7,29]. However, Eteläinen et al. showed that intracellular ROS levels reacted to KYP-2047 in a dose–response manner, and some cell lines required higher concentrations of KYP-2047 than we studied before the antioxidant effect of KYP became statistically significant [7]. The trend toward reduced ROS levels that we observed in our cells is in line with their reports on SH-SY5Y cells, which showed a trend toward reduced ROS at 1 µM KYP-2047 and became significant only at 10 µM [7]. Further studies with higher concentrations of KYP-2047 could reveal its anti-oxidative character more specifically. Here, we focused on analyzing the lowest cytoprotective concentration of KYP-2047.

Increased PREP activities have been linked to inflammatory conditions [6,8,10], and PREP is known to regulate the production of specific inflammatory mediators. PREP is a major regulator of acetylseryl-aspartyl-lysyl-proline in rat kidneys (Ac-SDKP), a tetrapeptide with anti-inflammatory, anti-fibrotic, and pro-angiogenic properties [30,31]. The production of the proinflammatory extracellular collagen-derived peptide N-acetylated prolyl-glycyl-proline (N-α-PGP) in cystic fibrosis samples clearly correlated with the PREP activity [6]. N-α-PGP is a homologue to the CXC-chemokines containing a Glu-Leu-Arg (ERL) motif (such as IL-8) and acts as a neutrophil chemoattractant [6]. Neutrophils also express PREP and are therefore able to generate N-α-PGP from collagen and thus self-sustain neutrophil-associated inflammation [32]. Penttinen et al. suggested that the regulation of inflammatory mediators by PREP could be the cause of its effect on memory loss, neurodegeneration, and cell survival [8].

Here, MG-132-mediated proteasome inhibition reduced the release of proinflammatory cytokine IL-6 but increased the production of proinflammatory cytokines IL-8 and IL-1β, which is consistent with our previous studies [21]. Similar results have been reported with several other cell types, such as human airway smooth muscle cells [33], oral squamous cell carcinoma-derived cell lines [34], and the 293T cell line [35]. In contrast, Qin and Gao demonstrated increased IL-6 secretion upon an exposure of ARPE-19 cells to MG-132 [36]. In their study, a higher concentration (10 µM) of MG-132 was used, which may explain their different findings in comparison with our study. In our cell cultures, the IL-6 levels were further reduced by the exposure of ARPE-19 cells to KYP-2047, indicating an anti-inflammatory effect for the PREP inhibitor. Even though KYP-2047 treatment tended to reduce the levels of both IL-8 and IL-1β, this decrease was not statistically significant. Natunen et al. observed reduced levels for the pro-inflammatory cytokine tumor necrosis factor alpha in the neuron-BV-2 neuroinflammation model after exposure of the cells to KYP-2047, a result in support of the anti-inflammatory effects observed in the current study [18].

JNK and other MAPKs are known regulators of inflammation [37]. Svarcbahs et al. showed that JNK phosphorylation was reduced by the PREP inhibitor KYP-2047 in HEK-293 cells and in the PREPko mouse cortex [22]. However, in contrast to the findings of Svarcbahs et al., JNK phosphorylation was not affected by KYP-2047 treatment in the ARPE-19 cells. The concentration of 1 µM KYP-2047 increased the phosphorylation of ERK1/2 and p38, indicating that the observed anti-inflammatory response of KYP-2047 in the ARPE-19 cells could be mediated via these pathways. To verify this hypothesis, the levels of IL-6 and IL-8 were measured in the presence of p38 and ERK1/2 inhibitors. Inhibition of p38 in the KYP-2047-treated cells resulted in increased production of IL-6 and IL-8, suggesting that the anti-inflammatory effect of KYP-2047 was being mediated via p38. This is in line with the effects of MG-132 exposure, which increased p38 activity and already reduced the IL-6 levels by itself. MG-132 is known to inhibit NF-κB, which might account for its inhibitory effect on IL-6 secretion, but in preliminary experiments, the effect of KYP-2047 did not appear to be linked to the NF-κB activity. We did not observe any increase in cytokine production upon ERK1/2 inhibition, suggesting that the phosphorylation of ERK1/2 was unrelated to the detected anti-inflammatory effects of KYP-2047.

Activation of the p38 kinase cascade has been most often associated with the initiation of an inflammatory response, and this is the reason why p38 inhibitors have been proposed as potential anti-inflammatory compounds [38,39,40,41]. However, it is known that active p38 also regulates inflammation via negative feedback loops and is consequently able to suppress inflammation [38]. These feedback loops involve regulation of the anti-inflammatory cytokine IL-10, mRNA destabilizing factor tristetraprolin (TTP), MAPK phosphatase 1 (MKP-1), and transforming growth factor β-activated kinase 1 (TAK-1) [42]. Furthermore, IL-10 has been shown to suppress proinflammatory cytokine production, such as that of IL-6 [43,44]. Active p38 contributes to the accumulation of the inactive form of TTP, which in turn becomes activated concurrently with deactivation of p38. Subsequently, TTP targets several proinflammatory cytokines for degradation [45]. It is worth noting that p38 is also known to be involved in cellular survival mechanisms after oxidative stress [46]. We have previously shown that the inhibition of p38 in ARPE-19 cells in conditions of oxidative stress leads to increased inflammation and cytotoxicity [47]. Andrade et al. recently reported that the photoprotective effect of the galectin-3 protein on UVA-induced damage in RPE cells involved reduced oxidative stress, decreased levels of IL-6, and increased p38 phosphorylation [48]. These findings are in line with the current finding that the activation of p38 phosphorylation is involved in the cytoprotective and anti-inflammatory actions of KYP-2047 in ARPE-19 cells. Further studies to elucidate the role of MAPK signaling and IL-10 in the anti-inflammatory effects of KYP-2047 in RPE cells suffering from dysfunctional proteasomal clearance will shed more light on the pathways and possible benefits of PREP inhibition in AMD.

## 5. Conclusions

The PREP inhibitor KYP-2047 displayed cytoprotective and anti-inflammatory effects as well as anti-oxidative potential upon MG-132-induced proteasomal damage in ARPE-19 cells. Our results suggest that the p38 signaling pathway is involved in the observed anti-inflammatory effects of KYP-2047.

## Figures and Tables

**Figure 1 antioxidants-12-01279-f001:**
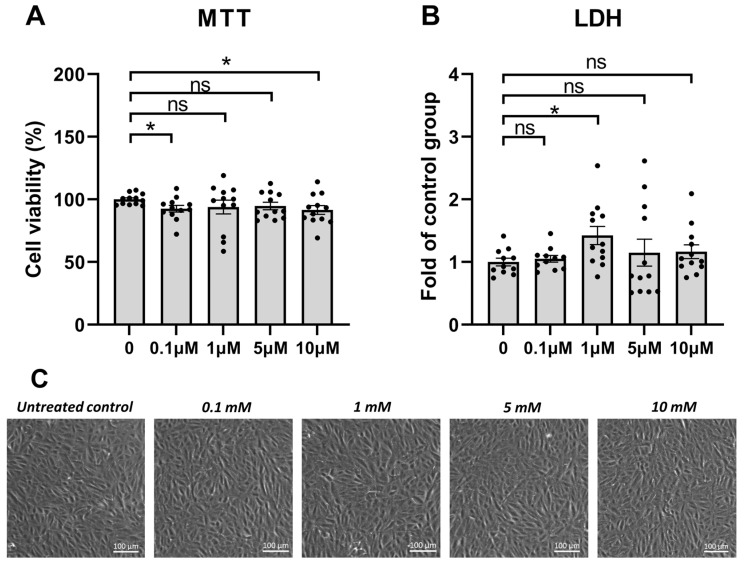
Cell viability of ARPE-19 cells upon 24 h of KYP-2047 exposure. The cells were exposed to different KYP-2047 concentrations (0.1, 1, 5, or 10 µM). Results were compared to the untreated control group, which was set to be 100% (**A**) or 1 (**B**). Data were combined from three experiments containing four parallel samples in each group per experiment (n = 12). Results are presented as bar plots with mean ± standard error of mean (SEM). Individual sample values are visible as black dots. Representative microscopy images are shown (**C**). * *p* < 0.05, ns = not statistically significant (Mann–Whitney U test).

**Figure 2 antioxidants-12-01279-f002:**
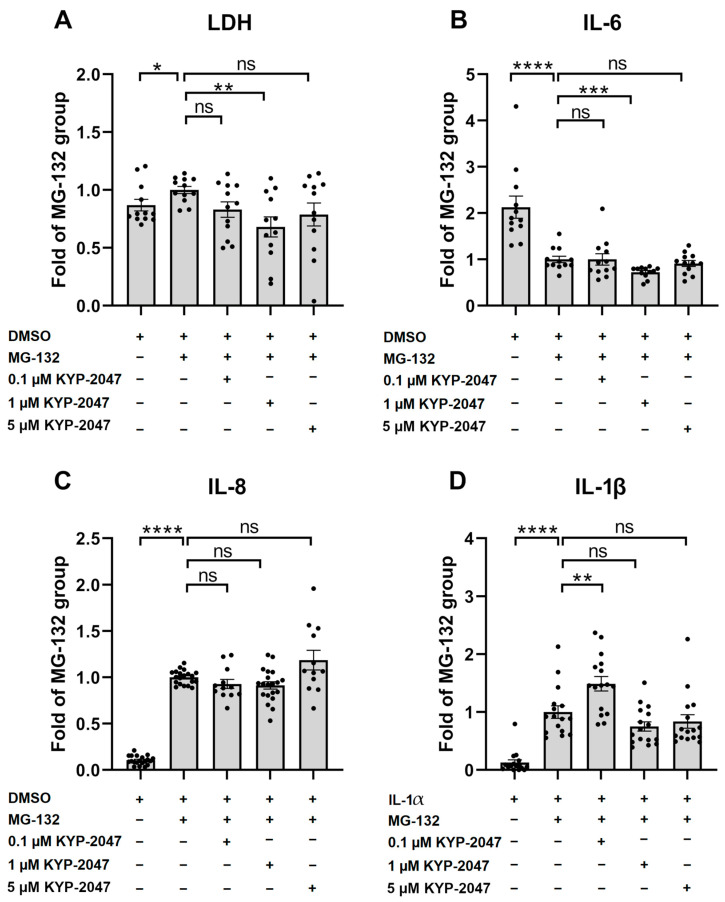
Effect of KYP-2047 on the release of LDH (**A**) upon proteasomal inhibition induced by 24 h of exposure to 5 µM MG-132, as well as the levels of proinflammatory cytokines IL-6 (**B**) and IL-8 (**C**). IL-1β (**D**) levels were measured after initial 24 h of priming with IL-1α and 48 h of exposure to KYP-2047 and MG-132. Results were compared to the MG-132 group, which was set to be one. KYP-2047 and MG-132 were dissolved and diluted in DMSO, and an identical amount of DMSO served as a vehicle control. LDH (**A**) and IL-6 (**B**) data were combined from three experiments containing four parallel samples in each group per experiment (n *=* 12). IL-8 (**C**) data were combined from five experiments (groups: DMSO, MG-132, and 1 µM KYP-2047 + MG-132; n *=* 20) and three experiments (groups: 0.1 µM KYP-2047 + MG-132 and 5 µM + MG-132; n *=* 12) containing four parallel samples in each group per experiment. IL-1β (**D**) data were combined from four experiments containing four parallel samples in each group per experiment (n *=* 16). Results are presented as bar plots with mean ± SEM. Individual sample values are visible as black dots. * *p* < 0.05, ** *p* < 0.01, *** *p* < 0.001, and **** *p* < 0.0001, ns = not statistically significant (Mann–Whitney U test).

**Figure 3 antioxidants-12-01279-f003:**
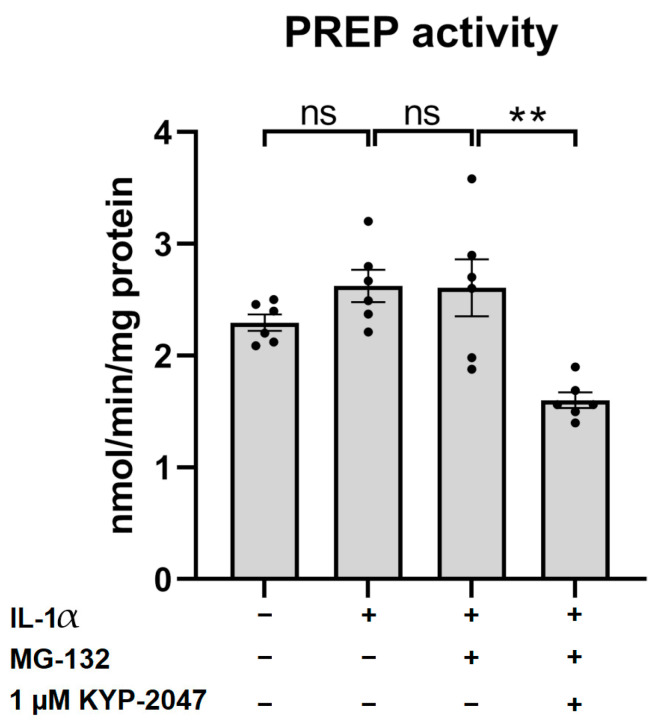
PREP activity 48 h after exposure to 5 µM MG-132 and 1 µM KYP-2047 in IL-1α-primed (4 ng/mL for 24 h) ARPE-19 cell lysates. Data were normalized to total protein concentrations. Results were combined from three experiments containing two parallel samples in each group per experiment (n *=* 6). Results are presented as bar plots with mean ± SEM. Individual sample values are visible as black dots. ** *p* < 0.01, ns = not statistically significant (Mann–Whitney U test).

**Figure 4 antioxidants-12-01279-f004:**
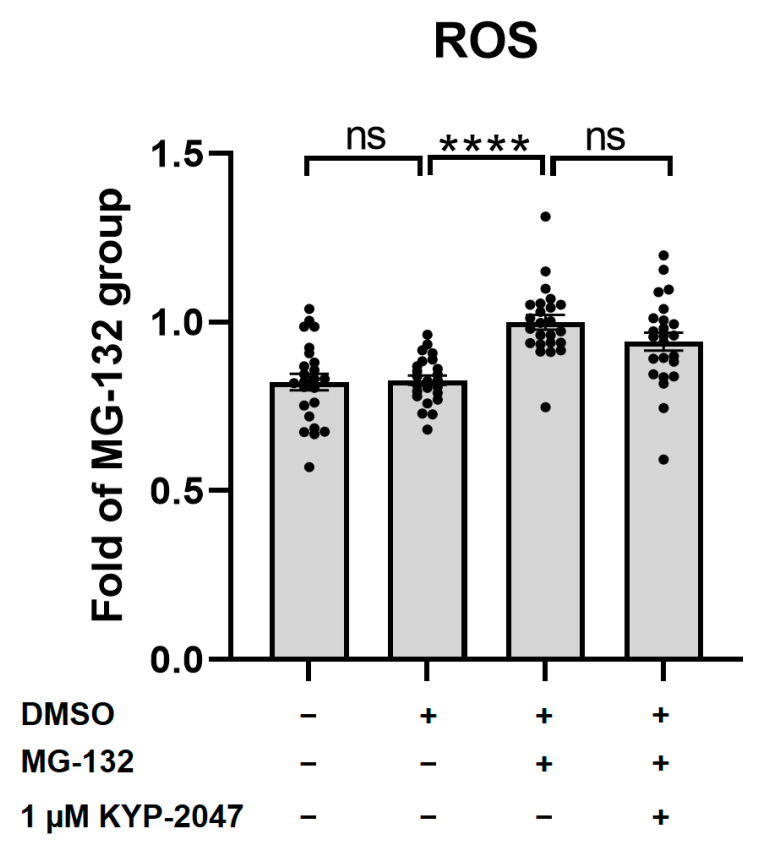
ROS production after IL-1α priming (4 ng/mL for 24 h), 15 min of incubation with 1 µM KYP-2047, as well as 5 µM H2DCFDA prior to 45 min of incubation with 5 µM MG-132. The exposure to MG-132 increased ROS production, and the PREP inhibitor tended to prevent it. Results were compared to the MG-132 group, which was set to be one. Data were combined from four experiments containing six parallel samples in each group per experiment (n *=* 24). Results are presented as bar plots with mean ± SEM. Individual sample values are visible as black dots. **** *p* < 0.0001, ns = not statistically significant (Mann–Whitney U test).

**Figure 5 antioxidants-12-01279-f005:**
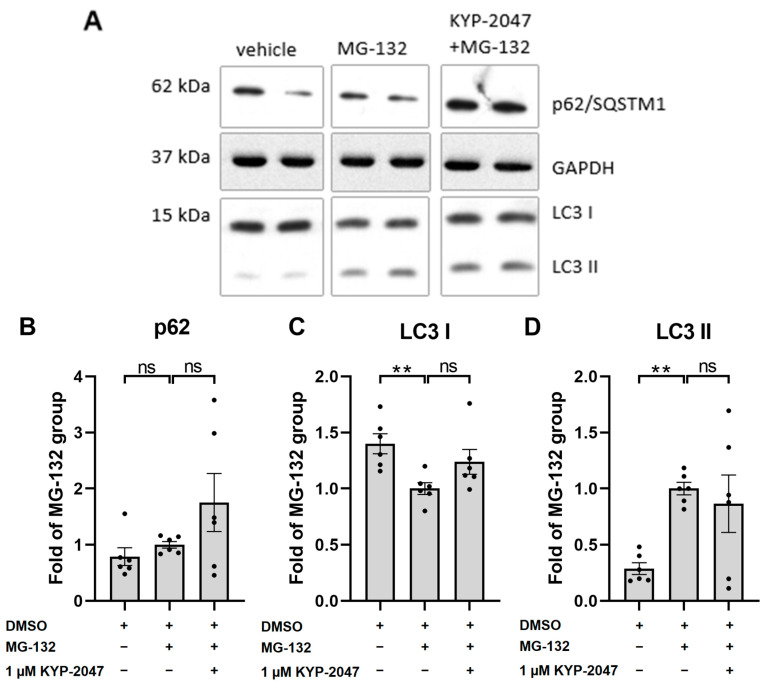
The levels of autophagy markers p62/SQSTM1 (**B**) and LC3 (**C**,**D**) 6 h after exposure to MG-132 and KYP-2047 were studied using the western blot method (**A**). Results were compared to the MG-132 group, which was set to be one. Data were combined from three experiments containing two parallel samples in each group per experiment (n *=* 6). Results are presented as bar plots with mean ± SEM. Individual sample values are visible as black dots. ** *p* < 0.01, ns = not statistically significant (Mann–Whitney U test).

**Figure 6 antioxidants-12-01279-f006:**
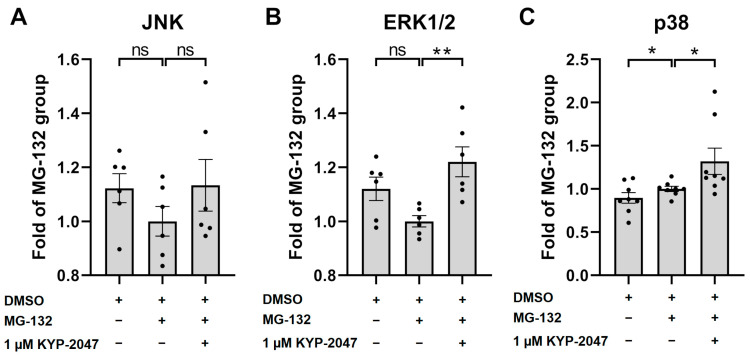
The levels of phosphorylated MAP kinases ERK1/2, JNK, and p38 after 2 h of exposure to MG-132 and KYP-2047. Results were compared to the MG-132 group, which was set to be one. Data were combined from three (**A**,**B**) or four (**C**) experiments containing two parallel samples in each group per experiment [n *=* 6 (**A**,**B**), n *=* 8 (**C**)]. Results are presented as bar plots with mean ± SEM. Individual sample values are visible as black dots. * *p* < 0.05. ** *p* < 0.01, ns = not statistically significant (Mann–Whitney U test).

**Figure 7 antioxidants-12-01279-f007:**
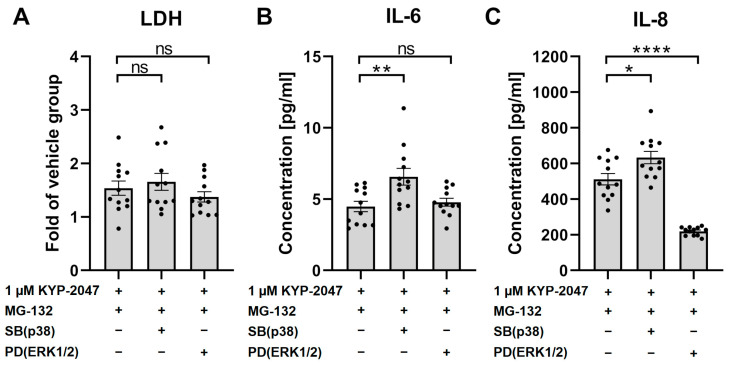
The levels of LDH (**A**), IL-6 (**B**), and IL-8 (**C**) after exposure to KYP-2047, MG-132, and p38 inhibitor SB or ERK1/2 inhibitor PD. The cells were exposed to KYP-2047 and SB or PD 15 min prior to 24 h of exposure to MG-132. LDH data were compared to the vehicle group, which was set to be one. Data were combined from three experiments containing four parallel samples in each group per experiment (n *=* 12). Results are presented as bar plots with mean ± SEM. Individual sample values are visible as black dots. * *p* < 0.05. ** *p* < 0.01. **** *p* < 0.0001, ns = not statistically significant (Mann–Whitney U test).

## Data Availability

The data presented in this study are available upon request from the corresponding authors.

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
