# Peer review of "The Prolyl Oligopeptidase Inhibitor KYP-2047 Is Cytoprotective and Anti-Inflammatory in Human Retinal Pigment Epithelial Cells with Defective Proteasomal Clearance"

_antioxidants, 2023, doi:10.3390/antiox12061279_

Round 1

Reviewer 1 Report

1.      The results shown in Fig 1B are confusing as the 1 uM treatment appears to be significantly toxic, while higher concentrations do not show any toxicity. It would be helpful if the author could provide a potential explanation for this observation.

2.      In Fig 2D, the authors do not include 0.1 uM KYP treatment. It would be helpful if the authors could provide an explanation for why this concentration was excluded from the study.

3.      In Fig 3, the author does not specify the treatment concentration of IL-1a and MG-132. It would be helpful to provide this information to allow readers to better understand the experimental conditions.

4.      There is no non-treatment negative control shown in Fig 4. It would be useful to include such a control to ensure that any observed effects are due to the treatment and not some other factor.

5.      The letterboxed western blot results shown in Fig 5A are difficult to interpret. It would be helpful if the authors could run all the samples on the same gel and provide the number of samples analyzed to improve the clarity of the results.

6.      In Fig 6, it would be useful if the authors could include the original western blot images used for the statistical analysis to allow readers to better evaluate the results.

7.      The experiment design shown in Fig 7 appears to lack proper controls. It would be helpful if the authors could provide additional information about the experimental design to clarify the interpretation of the results.

Reviewer 2 Report

The manuscript entitled "The prolyl oligopeptidase inhibitor KYP-2047 is cytoprotective and anti-inflammatory in human retinal pigment epithelial cells with defective proteasomal clearance" and submitted for publication in Antioxidants, addresses an interesting topic, AMD, is well written and presents a set of solid results.

However, there are a number of aspects and doubts that need to be clarified.

Thus:

1) The authors use in this work the ARPE-19 cell line, a widely used RPE model. These cells have a number of characteristics that pigmented epithelial cells lose as the number of passages increases. According to the authors, the cells used in this work had between 26 and 35 passages, which is already a relatively high number. Could the results obtained be due, at least partially, to the fact that the cells are already in a marked process of differentiation and mesenchymal transition? Why not perform some of the experiments presented with cells with a much lower number of passages?

2) In figure 1 the authors present the results of incubating the cells with different concentrations of KYP-2047, showing that 5 uM does not induce toxicity, unlike 1 uM that induced an increase in LHD leakage. However, in figure 2 the authors only tested the effects of 0.1 and 1 uM by not testing 5 uM. What is the justification for not testing this higher concentration that has not been shown to have toxic effects?

3) In figure 3 the authors show the effect of the inhibitor KYP-2047 on PREP activity. However, the addition of 1uM of KYP-2047 only partially inhibited PREP, and the activity was quite high (70% of the control). Couldn't this activity of PREP, even in the presence of the inhibitor, make it difficult to interpret the results? What will happen to the activity with higher contractions of the inhibitor (5uM)?

4) What are the protein levels of PREP in the various experimental conditions? (Control, with MG-132 and with MG+KY)

5) The show that MG-132-mediated proteasome inhibition reduced the release of proinflammatory cytokine IL-6 from ARPE-19 cells. However, a 2018 study (https://www.hindawi.com/journals/joph/2018/5392432/) showed that MG-132 had an opposite effect, increasing IL-6 levels in the same cells. Authors should include this work in their references and seek to justify this apparent contradiction.

6) Authors should choose to present the results in Scatter plot graphs with bars rather than column graphs, so readers can better visualize the results.

Round 2

Reviewer 2 Report

In the revised version of the manuscript, the authors have taken into account many of the suggestions made, so that I now consider that this work meets the requirements to be accepted for publication in Antoxidants.

Author Response

We thank the reviewer for this positive statement.